# Association of Alpha B-Crystallin Expression with Tumor Differentiation Grade in Colorectal Cancer Patients

**DOI:** 10.3390/diagnostics11050896

**Published:** 2021-05-18

**Authors:** Cristina Pagano, Giovanna Navarra, Patrizia Gazzerro, Mario Vitale, Maria Notarnicola, Maria Gabriella Caruso, Elisabetta Cavalcanti, Raffaele Armentano, Chiara Laezza, Maurizio Bifulco

**Affiliations:** 1Department of Molecular Medicine and Medical Biotechnologies, University of Naples “Federico II”, 80131 Naples, Italy; pagano.cris@gmail.com (C.P.); vanna.navarra@libero.it (G.N.); 2Department of Pharmacy, University of Salerno, 84084 Fisciano, Salerno, Italy; pgazzerro@unisa.it; 3Department of Medicine, Surgery and Dentistry “Scuola Medica Salernitana”, University of Salerno, 84084 Salerno, Italy; mavitale@unisa.it; 4Laboratory of Nutritional Biochemistry, National Institute of Gastroenterology, “S. de Bellis” Research Hospital, Via Turi 27, 70013 Castellana Grotte, Italy; maria.notarnicola@irccsdebellis.it; 5Clinical Nutrition Outpatients Clinic, National Institute of Gastroenterology ‘S de Bellis’ Research Hospital, Via Turi 27, 70013 Castellana Grotte, Italy; gabriella.caruso@libero.it; 6Research Department, National Institute of Gastroenterology “S. de Bellis” Research Hospital, 70013 Castellana Grotte, Italy; elisabetta.cavalcanti@irccsdebellis.it (E.C.); raffaele.armentano@irccsdebellis.it (R.A.); 7Institute of Endocrinology and Experimental Oncology G. Salvatore, CNR, 80131 Naples, Italy; chilaez@hotmail.com

**Keywords:** Alpha-β-crystallin, colorectal adenocarcinoma, hystological grade, tissues samples

## Abstract

Alpha B-crystallin (CRYAB, HSPB5) belongs to the small heat shock protein (HSP) family and is highly expressed in various human cancers, suggesting a crucial role in tumor progression. However, few studies have examined CRYAB expression in colorectal cancer (CRC). In the present study, we investigated the relationship between CRYAB expression and the clinicopathological features of CRC samples. We comparatively analyzed CRYAB protein expression in 111 CRC tissues and normal adjacent colonic tissue, observing that it was significantly lower in CRC tissues than in corresponding non-cancerous tissues. Moreover, immunohistochemical analysis showed a significant correlation between CRYAB expression and high histological grade G3 (*p* = 0.033). In summary, our results point to its possible application as a prognostic biomarker in CRC patients.

## 1. Introduction

Colorectal cancer (CRC) is the most common cancer worldwide, with a poor prognosis. The 5-year survival rate is ~45% despite progress in early diagnosis and advanced therapeutics. In order to improve screening, therapeutic treatment, and outcomes of CRC patients, the detection of new biomarkers is becoming a crucial goal in clinical oncology [1]. Approximately 15% of all CRCs are screened for DNA mismatch repair/microsatellite status characterized by microsatellite instability (MSI), indicating a defect in mismatch repair (MMR). At present, the only other molecular marker qualified for CRC prognosis is KRAS/BRAF mutational analysis, which can be used to select the most effective treatment of CRC patients. At present, the most widely studied prognostic factors are clinical and pathologic features including tumor size, differentiation grade, TNM stage, and metastasis status [2,3]. Alpha B-crystallin (CRYAB or HspB5) gene is located in the tumor-suppressive 11q22-23 region. Besides the crucial role for vision in retinal cells, CRYAB is a molecular chaperone that inhibits the aggregation of unfolded proteins, preventing their degradation caused by cellular stress stimuli such as heat shock, oxidative stress, or radiation [4], thereby promoting cell survival and preserving cytoskeletal integrity [5,6]. Indeed, CRYAB malfunction has been associated with myopathy, neuropathy, ischemia, cataract, and cancer [7]. It has been observed that CRYAB protects against apoptosis interacting directly with caspase-3, Bax, and Bcl-xS, thus hindering their translocation to the mitochondria. CRYAB also inhibits apoptosis mediated by Raf/MEK/ERK signaling pathway by preventing RAS activation. In other studies, it has been described that it inhibits apoptosis by activation of Akt signaling pathway and PI3K activity [8]. CRYAB high expression induces EMT and metastasis by NF-κB signaling pathway in gastric cancer [9], and in human basal-like breast tumors, the MAPK kinase/ERK1/2 pathway [10]. Down-regulation of CRYAB expression is associated with nasopharyngeal [11], testicular [12], and breast [13] cancers, suggesting that CRYAB acts as a tumor-suppressor gene. CRYAB over-expression was detected in mammary metaplastic carcinomas promoting brain metastasis in breast cancer, while silencing the CRYAB gene inhibited distant metastasis [8]. Various clinical studies have highlighted that high CRYAB expression is a prognostic biomarker for various human cancers including CRC [14]. However, the relationship between CRYAB expression and the clinical/pathological features of CRC has been rarely evaluated. Whether CRYAB also acts as an oncogene or whether CRYAB could be a valuable biomarker in CRC might have great clinical significance and help to understand the process of CRC progression.

## 2. Material and Methods

### 2.1. Clinical Samples

One hundred and eleven formalin-fixed and paraffin-embedded tissue specimens were collected from patients underwent surgery for colorectal cancer at the Surgery Division of our institute. These patients were enrolled in the context of the study approved by the Ethical Committee of IRCCS “S. de Bellis”, Castellana Grotte (Bari, Italy, number code: 32/CE/DE BELLIS, 27 October 2016). The study was conducted in accordance with the Declaration of Helsinki and all patients were invited to provide an informed consent to take part in the study. At surgery, samples of mucosa, taken from macroscopically normal areas of intestine, at 10 cm from the neoplasia and cancer tissue, were obtained for each subject and stored at −80 °C until assayed.

For all patients, we collected the following clinicopathological features: age, gender, primary site, tumor grade, and tumor node metastasis (TNM).

### 2.2. Immunohistochemistry (IHC)

IHC analysis for CRYAB protein was performed in tumor specimens selected on the basis of hematoxylin–eosin (HE) staining. Tumor sections of 4 μm were freshly cut and dried at 60 °C for 30 min. IHC analysis was carried out in sections after deparaffinization for 30 min and then dehydrated in alcohol. Antigen retrieval was performed at 90 °C for 20 min with citrate buffer. CRYAB protein staining was obtained by specific antibodies clone F-10 (Santa Cruz Biotechnology, Inc., CA, USA) at 1:100 dilution using an automated autostainer (cat. K5007, Dako, Glostrup, Denmark). The Real Envision DAB Substrate Kit (DAKO) was used according to the manufacturer’s instructions. Finally, the samples were observed under a light microscope (Olympus Corp, Tokyo, Japan).

### 2.3. IHC Assessment

CRYAB was expressed both inside the cells and on the cellular membrane, mainly seen on the internal surface of the tumor cell membrane. Averages of intensity and percentage of positively stained cells were assessed by three independent observers. The intensity of staining was scored as 0 (no staining), 1 (weak staining = light yellow), 2 (moderate staining = yellow-brown), and 3 (strong staining = brown). The percentage of stained cells was scored as 0 (no positive cells), 1 (less than 25% positive cells), 2 (25–50% positive cells), 3 (more than 50–75% positive cells), and 4 (more than 75% positive cells). Scoring was validated by a consulting histopathologist. Positive staining was judged by the presence of an unequivocal brown staining in ≥10% of tumor cells.

### 2.4. Western Blot Analysis

On the basis of the availability of frozen tissue samples stored at –80 °C, we performed Western blot analysis in 49 of 111 patients. Total protein extracts were obtained treating each tissue sample with total lysis buffer (Pierce Ripa buffer, Thermo Scientific, Rockford, IL, USA) supplemented with protease and phosphatase inhibitors (Thermo Scientific, Rockford, IL, USA). After homogenization and centrifugation at 14,000 rpm for 15 min at 4 °C, the protein concentration was measured by a standard Bradford assay (Bio-Rad, Milan, Italy). Aliquots of 50 µg of total protein extracts from each sample were denaturated in 5 × Laemmli sample buffer and loaded into 4–12% pre-cast polyacrylamide gels (Bio-Rad, Milan, Italy) for Western blot analysis. CRYAB antibody (clone F-10 Santa Cruz Biotechnology, Inc., CA, USA) and beta-actin (Santa Cruz) were used as primary antibodies. After overnight incubation, the membranes were incubated with a horseradish peroxidase-conjugated secondary antibody (Bio-Rad, Milan, Italy). The proteins were detected by chemiluminescence (ECL, Thermo Scientific, Rockford, IL, USA), and each protein-related signal was obtained using the Molecular Imager Chemidoc^TM^ (Bio-Rad, Milan, Italy) and normalized against beta-actin protein expression.

### 2.5. Statistical Analysis

A chi-squared test of cross-tabulations and Fisher’s exact test were used to examine the relationship between the expression of CRYAB and the clinicopathologic characteristics of patients. We used the Wilcoxon signed-rank test to explore the different expressions of CRYAB. Data are presented as the mean ± SD. Statistical analyses were performed using GraphPad Prism (Graphpad Software 7.0; GraphPad, San Diego, CA, USA). The Kaplan–Meier Survival and log-rank test method were used to estimate, respectively, the difference of overall survival (OS) and disease-free survival (DFS) between high- and low-CRYAB TPM, as well as to evaluate the equality of survival among categories. The Cox model is a statistical technique for exploring the relationship between the survival of a patient and singular or several explanatory variables, and also it allowed us to estimate the hazard risk (HR) of survival for an individual, given their prognostic variables (measured as continuous or categorical). The Cox proportional hazard model was fitted to the data, and the proportional hazard assumption was evaluated by means of Schoenfeld residuals (SRT). All models for fitting were evaluated by means of Akaike information criteria (AIC) and Bayesian information criterion (BIC). Risk estimators were expressed as hazard ratios (HRs) and 95% confidence interval (95% CI). In the models, the multicollinearity was evaluated through the variance inflation factor (VIF), using a score of 2 as cut-off for exclusion.

## 3. Results

### Expression of CRYAB Protein

CRYAB protein expression was investigated in 111 paraffin-embedded archived CRC samples by performing immunohistochemistry with an antibody against human CRYAB (Table 1 and Table 2). Twenty-three CRC samples were negative with absent staining, whereas 88 samples showed detectable immunoreactivity (weak, intermediate, strong). The immunoreactivity of CRYAB was present mainly in the cytoplasm of the tumor cells. In well-differentiated colon adenocarcinoma (G1), CRYAB protein was present in the muscolaris mucosae, in the neural structures, in the interstitial cells of Cajal, and in the intramural nerve plexuses (Figure 1B,C), while it was absent in the neoplastic glandular structures. In moderately differentiated colon adenocarcinoma (G2), CRYAB protein was intensely stained in tumor cells (>20%) (Figure 1D–F). In the poorly differentiated colon adenocarcinoma (G3), CRYAB expression was negative in tumor cells (Figure 1G,H), while it was evident in the interstitial cells of Cajal, in the intramural plexuses, and in the leiomuscular residues of the intestinal wall. The staining was expressed exclusively in the G2 component (moderately differentiated) compared to G1 and G3 samples (Figure 1). Statistical analyses were carried out to investigate the correlation between expression and the clinicopathological features of CRC. As shown in Table 1, no significant correlation was found between the expression level and gender, age, tumor differentiation, tumor site, or tumor size. Of note, the expression of CRYAB was correlated with the differentiation stage of patients with CRC (*p* = 0.033) Table 2. Figure 2 shows the Western blot data performed in 49 CRC tissues and corresponding adjacent non-cancerous mucosa. The representative immunoblot showed a different protein levels in the tumor tissues depending on histological grade of CRC. In the poorly differentiated colon adenocarcinoma (G3) as well as in well-differentiated colon adenocarcinoma (G1), CRYAB was poorly expressed in the tumor tissues compared with the adjacent normal mucosa, while in moderately differentiated tissues (G2), tumor tissues showed significantly higher CRYAB protein levels compared with the adjacent normal mucosa (Figure 2). The histogram represents the mean relative expression of the tumor tissues of 49 CRC patients analyzed. The measurement of O.D. absorption of WB was 25.52 ± 10.5 and 11.4 ± 17.7 for normal mucosa and tumor tissue, respectively (mean ± S.D., *p* < 0.0001). This result was in accordance with that obtained by GEPIA interactive web server [15] on the dataset of The Cancer Genome Atlas Program (TCGA) project (https://www.cancer.gov/tcga, accessed on 1 February 2021) (Figure 3A). We also reported the overall survival (OS) and disease-free survival (DFS) datasets of TCGA, indicating that patients with high expression of CRYAB TPM (red line) had significantly worse survival than those with low expression of CRYAB TPM (blue line) (*p* = 0.0058 for OS and *p* = 0.0093 for DFS by log rank test. The high CRYAB TPM category had a median time of 60 days about for OS and DFS, while patients in low CRYAB TPM category survived. Regarding hazard risk (HR) in each analysis, we had an HR > 1 (HR = 2, *p* = 0.0068 for OS, and HR = 1.9, *p* = 0.011 for DFS), indicating that high CRYAB TPM category had a negative effect on patient survival with a risk of 2-fold greater than of low CRYAB TPM. (Figure 3 B–C)

## 4. Discussion

While a large number of studies describes a relationship between CRYAB expression and different kinds of cancers, only few consider colon cancer. CRYAB over-expression has been described to be correlated with poor prognosis in breast carcinoma, head and neck cancer, and lung and hepatocellular carcinoma. In CRC patients, the elevated expression in lymph node metastasis, distant metastasis, and tumor TNM stage have been found to be associated with the overall survival. Moreover, the high expression of CRYAB could promote proliferation, invasion, and metastasis of CRC through EMT [8].

In the current study, including 111 patients with colonic cancer of different histological grade, we investigated the CRYAB differential expression in between CRC and matched normal tissues. The results demonstrated that the protein expression of CRYAB was downregulated in 49 CRC tissues compared with matched non-cancerous tissues. In addition, we highlighted that low CRYAB protein expression was correlated with certain clinicopathological parameters, including hystological grade G3. Similarly, the data of IHC analysis in 111 CRC samples also indicated a lower expression of CRYAB protein in CRC samples of hystological grade G3 compared to non-cancerous tissues and CRC tissues with G2 component (moderately differentiated).

These data suggest that CRYAB expression was significatively associated with the tumor grade, meaning that lower expression of the protein is found in poorly differentiated tumors (G3) (χ^2^ test, *p* = 0.033). However, there was no correlation with gender, primary site, or lymph node status. Moreover, our study demonstrated a significant association with the grade and CRYAB expression levels in CRC tissues. In particular, G3 tumors were characterized by lower CRYAB expression, reflecting a predisposition of colon cells to a less differentiated hyperproliferative state. Lastly, our results indicated that CRYAB might be a promising molecular marker for CRC treatment, which might facilitate the development of precise treatment selection and improve the outcome for CRC patients. Further research that enroll larger samples and elucidate the mechanisms of CRYAB action are necessary.

## 5. Conclusions

Colorectal cancer (CRC) is a major cause of morbidity and mortality in the world. Research on early diagnosis of biomarkers is crucial to select “personalized” treatment strategies to improve the prognosis of this disease. The present study has described the correlation of CRYAB expression with poorly differentiated adenocarcinoma of the colon (G3). The grading of cancer tissue samples is important for the prognosis of cancer patients and for the therapeutic decision of the clinician; for this reason, the CRYAB should be explored as a biomarker candidate for CRC diagnosis and prognosis. These results are of great relevance in future studies to implement this gene as prognostic factors.

## Figures and Tables

**Figure 1 diagnostics-11-00896-f001:**
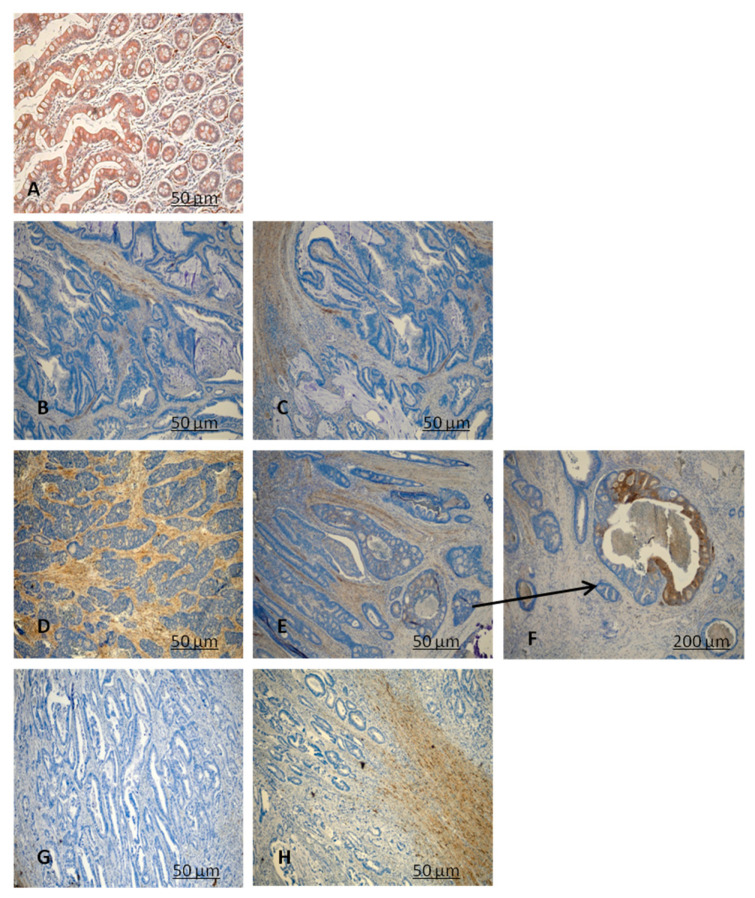
Representative figures of CRYAB protein expression in colorectal cancer (CRC) tissues (**B–H**) and non-cancerous tissues (**A**) by immunohistochemistry (IHC) analysis. (**A**) High expression of CRYAB in non-cancerous cells. (**B**,**C**) (well differentiated) low expression in the neoplastic glandular structures; (**D**–**F**) (moderately differentiated) high expression of CRYAB in neoplastic cells; (**G**,**H**) (poorly differentiated) negative expression of CRYAB in CRC cells. (**A**–**E**,**G**,**H**) 10×; (**F**) 40× magnification.

**Figure 2 diagnostics-11-00896-f002:**
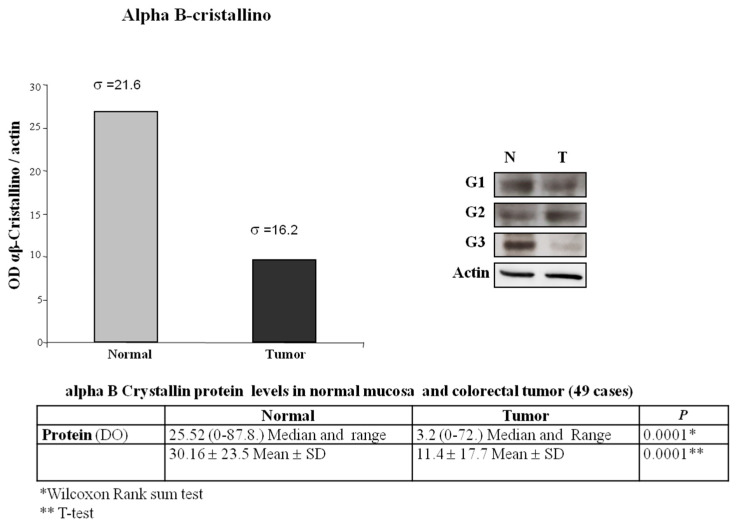
Optical density and Western blot of CRYAB protein expression in CRC tissues (T) and adjacent non-cancerous mucosa (N) from the same patient.

**Figure 3 diagnostics-11-00896-f003:**
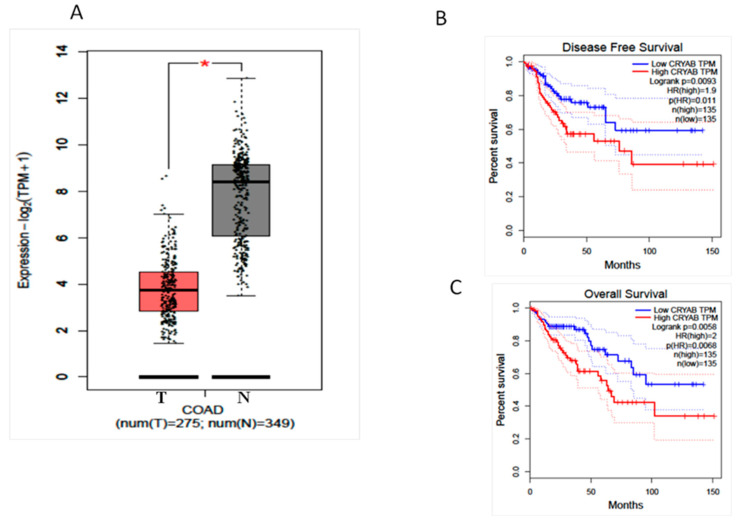
(**A**) CRYAB gene expression comparison between groups: COAD-TCGA, adjacent non-tumoral tissue. CRC: colorectal cancer; COAD: colon adenocarcinoma. Median is represented as a solid black line, whereas mean is represented as a black square. *p*-value cutoff: 0.01. (**B**) Disease-free survival (DFS) datasets of TCGA of (**C**) overall survival (OS) and datasets of TCGA.

**Table 1 diagnostics-11-00896-t001:** Correlation of CRYAB expression with clinicopathological characteristics of 111 CRC patients.

Group	Cases	CRYAB	
	*n* = 111	High	Low
**Gender**		*n* = Cases	*n* = Cases
Male	78	36	25
Female	32	22	17
**Age**			
>60	72	37	35
<60	39	21	18
**Hystological Type**			
Adenocarcinoma	100	30	70
Mucinoso	11	9	2
**Tumor Differentiation**			
**G1**	19	4	15
**G2**	59	31	28
**G3**	33	8	25
**TNM**			
Stage I–II	52	27	25
Stage III–IV	59	29	30

**Table 2 diagnostics-11-00896-t002:** Correlation of CRYAB positive cells with Hystopatological type and Grading of 111 CRC patients.

Valid Cases *n =* 111			
**Hystopatological type × % positive cells Cross tabulation**			
(2-sided)	Value	df	*p* value
Pearson Chi-Square	16.768	8	0.033
Likelihood Ratio	12.627	8	0.125
**GRADING × % positive cells Cross tabulation**			
Pearson Chi-Square	9.645	4	0.047
Likelihood Ratio	8.039	4	0.090

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
