# Peer review of "Association of Alpha B-Crystallin Expression with Tumor Differentiation Grade in Colorectal Cancer Patients"

_diagnostics, 2021, doi:10.3390/diagnostics11050896_

Round 1

Reviewer 1 Report

The author has modified the manuscript based on the comment. 

Reviewer 2 Report

The revised manuscript is now recommended for consideration for acceptance. Just check for the legend of figure 3. 

This manuscript is a resubmission of an earlier submission. The following is a list of the peer review reports and author responses from that submission.

Round 1

Reviewer 1 Report

The authors of the manuscript "Association of Alpha B-crystallin expression with tumor differentiation grade in colorectal cancer patients" evaluate the diagnostic and prognostic value of alpha-B crystallin expression in CRC tumors.

The authors claim a reduced protein expression of alpha-B crystallin in tumor versus normal tissue, with significant expression alterations according to histological type and grading. Although representative WB images are missing, the lower expression in CRC tumors is backed by publicly available transcriptome data. Although the presented data are in conflict with previous publications (Shi et al., Int J Clin Exp Pathol, 2014; Li et al., PLOSone, 2017) its significant differential expression in poorly differentiated tumors might improve patient stratification for adjuvant chemotherapy (Ueno et al., Am J Surg Pathol, 2020). Still, a lower expression of alpha-B crystallin in a tumor grade with higher relapse rate is conflicting with its reported impact on CRC progression and metastasis.

Reviewer 2 Report

This Brief report examined CRYAB expression in CRC and analyzed the association with the clinical features of CRC. Generally, the methods of this analysis are sound and the results is fully discussed, but there are some concerns need to address.

  1. The description of Tables is bad, the authors should listed the results as common three line format.
  2. The authors stated that there was Western-blot examination in the Method part, but I did not find the corresponding results in the Results part.
  3. In the conclusion, the authors stated that “a possible application as diagnostic and prognostic biomarker in CRC patients”, but in my opinion, these results is far from the association with diagnostic and prognostic value, it merely associated with the pathogenesis of CRC.

Reviewer 3 Report

The present paper has identified the CRYAB gene as a molecular marker in CRC patients. This study sheds new light on the association of  CRYAB expression with tumor differentiation grade in CRC patients. Certainly, this study is interesting and significant, however, there are some points which the authors need to be addressed.

Comments

  • The author should significantly expand the Introduction or background of CYRAB and associated signaling pathways.
  • Is the lower expression of CYRAB associated with Disease-Free Survival and Overall survival in colon adenocarcinoma from the available database?
  • The author should include the survival analysis of the G1, G2, and G3 groups.
  • Please provide the scale bar in the images.
  • The authors are encouraged to prepare the manuscript more carefully specially typo errors, figure numbers.